# Atomic electrostatic maps of 1D channels in 2D semiconductors using 4D scanning transmission electron microscopy

Shiang Fang[1], Yi Wen[2], Christopher S. Allen [2,3], Colin Ophus[4], Grace G.D. Han[5], Angus I. Kirkland [2,3], Efthimios Kaxiras[1,6] & Jamie H. Warner[2]

Defects in materials give rise to fluctuations in electrostatic fields that reflect the local charge density, but imaging this with single atom sensitivity is challenging. However, if possible, this provides information about the energetics of adatom binding, localized conduction channels, molecular functionality and their relationship to individual bonds. Here, ultrastable electron-optics are combined with a high-speed 2D electron detector to map electrostatic fields around individual atoms in 2D monolayers using 4D scanning transmission electron microscopy. Simultaneous imaging of the electric field, phase, annular dark field and the total charge in 2D $MoS_2$ and $WS_2$ is demonstrated for pristine areas and regions with 1D wires. The in-gap states in sulphur line vacancies cause 1D electron-rich channels that are mapped experimentally and confirmed using density functional theory calculations. We show how electrostatic fields are sensitive in defective areas to changes of atomic bonding and structural determination beyond conventional imaging.

[1] Department of Physics, Harvard University, Cambridge, MA 02138, USA. [2] Department of Materials, University of Oxford, 16 Parks Road, Oxford OX1 3PH, UK. [3] Electron Physical Sciences Imaging Center, Diamond Light Source Ltd., Didcot, Oxfordshire OX11 0DE, UK. [4] National Center for Electron Microscopy, Molecular Foundry, Lawrence Berkeley National Laboratory, 1 Cyclotron Road, Berkeley 94720 CA, USA. [5] Department of Chemistry, Brandeis University, Waltham 02453 MA, USA. [6] John A. Paulson School of Engineering and Applied Sciences, Harvard University, Cambridge, MA 02138, USA. These authors contributed equally: Shiang Fang, Yi Wen. These authors jointly supervised this work: Efthimios Kaxiras, Jamie H. Warner. Correspondence and requests for materials should be addressed to E.K. (email: kaxiras@physics.harvard.edu) or to J.H.W. (email: Jamie.warner@materials.ox.ac.uk)

**4D** scanning transmission electron microscopy (4D-STEM) is gaining momentum for probing materials at sub-Angstrom resolution with the full electron −atom scattering interactions recorded in a convergent beam electron diffraction pattern (CBED)[1–4]. This has been revolutionized by high-speed electron detectors, either in the form of 2D pixelated cameras (2D-PCs) or as segmented detectors[5,6]. These have enabled strain maps across samples with picometer precision, and deep sub-Angstrom spatial resolution using ptychographic reconstruction methods[7,8].

Direct collection of CBED patterns on 2D-PCs provides rich information about phase and momentum transfer from the electron beam interactions with the samples' electrostatic fields[9,10]. Phase data can be reconstructed using pytchographic methods, together with simultaneously recorded ADF-STEM images[11]. The intensity fluctuations of the CBED pattern are used to produce differential phase contrast images that relate to momentum transfer to the electron beam as it propagates through the samples' electrostatic fields[12,13]. Atomic resolution images of electrostatic fields and charge distributions have been recorded for bulk crystals, such as GaN, where beam damage does not limit the long acquisition times[3]. Using 2D-PCs, this is done by measuring the intensity center of mass, while for quadrant detectors, the differential signal between opposite quadrants is used. Translating 4D STEM to the single atom level is more difficult because of the rapid sample damage at time scales faster than the acquisition speed and hence low beam dose is essential[13]. Furthermore, mapping features around single atoms in defects is challenging due to the low signal to noise[14,15]. However, 2D materials do offer a thin volume for direct interpretation in electron microscopy[16–18], and to study fluctuations of electrostatics around single atoms.

For semiconducting 2D monolayers, transition metal dichalcogenides (TMDs), such as $MoS_2$ and $WS_2$, form ultralong 1D channels by S sputtering at high temperature[19]. Density functional theory (DFT) calculations show that as the width of the S line vacancies increases from 1S to 2S, the band gap narrows from 1.9 to <0.1 eV, and becomes metallic at 4S width[20]. Theory suggests that these 1D conduction channels are due to the metal-rich bonding areas that form within the larger vacancy sections, but experimental verification of this has yet to be achieved with sufficient resolution to identify charge variations in regions of single atomic bonds. These W−W bonds create 1D sub-nm conduction channels in the 2D semiconductors with potential use in nanoscale electronics and devices. However, the detailed atomic structure of the ultralong 2S and 3S line vacancies is complex and difficult to accurately determine using only ADF-STEM or phase contrast images. Therefore, the multicomponent images obtained from 4D STEM, including total charge maps, are crucial to gaining a better understanding of the structure −property correlations. Furthermore, by using first principle calculations, we can determine the predicted electric fields and total charge values in these monolayer systems and quantitatively compare it to the experimental values. Prior work has primarily used image simulation methods to compare to experimental 4D STEM results.

Here, we show that 4D STEM can directly image electrostatic fields, total charge and phase maps with atomic resolution in monolayer $MoS_2$ and $WS_2$ 2D crystals with qualitative agreement to the predicted values from DFT calculations. Experimental values are quantitatively half of the DFT predicted values and this stimulates further investigations. Measurements are performed on sulfur line vacancies that form 1D channels at high temperature through vacancy diffusion into ordered lines. Metal−metal bonding is present in the S line vacancies and is shown to lead to electron-rich channels that act as in-gap states for 1D conduction.

More complex line vacancies with wider S vacancy regions are studied and show significant modulation of electric fields around atoms. Using a combination of ADF-STEM, phase imaging, electric field and total charge images, we are able to deduce the atomic structure of complex defective regions with a higher degree of certainty than using just one form of imaging contrast alone. The high sensitivity of the electric field maps to atomic bond coordination provides spatial information about nearest neighbor atoms that is not easily extracted from ADF-STEM images or phase maps.

## Results

**4D STEM of pristine 2D $MoS_2$ and $WS_2$ monolayers**. Figure 1a shows a diagram of the 4D STEM geometry used, where TMDs are suspended across holes within an in situ heating holder, a 60 kV electron beam is used to minimize damage and a high-speed 2D-PC is used to capture CBED patterns at each scan position (Fig. 1b), as well as low angle ADF-STEM (LAADF-STEM) data. A separate annular detector records the high-angle ADF-STEM (HAADF-STEM) image and provides complimentary information. Each 4D STEM scan takes ~2 min, and requires extreme sample and lens stability, minimal contamination and an electron dose sufficiently low to minimize beam-induced damage. Different probe positions show variations in the CBED pattern (Fig. 1b–d). The electric field perpendicular to the electron beam direction ($E⊥$) is proportional to the averaged quantum mechanical expectation value for momentum, which is measured from the center of mass ($I_{CoM}$) of the CBED intensity distribution[10]. The high stability of the 4D STEM setup is demonstrated by the HAADF image (Fig. 1e) recorded simultaneously with the 4D STEM data, showing minimal optical and sample drift.

The $x$ and $y$ components of the experimental $I_{CoM}$ can be used to calculate the momentum transfer and then the probe convoluted electric field components ($Ex⊥$ and $Ey⊥$). These are directly compared to the DFT calculated values and we typically find the experimental values being half that of the DFT. A prior report that compared the experimental electric field values to image simulations found the experimental values to be half and suggested that residual aberrations, partial coherence, and defocus may be the cause[10,18]. Here we used a scaling factor of ~2 to the 4D STEM data to achieve a match to the DFT-calculated maps (Fig. 1f–i, supplementary figures 1 and 2). The total magnitude of the probe convoluted electric field $E⊥$ is calculated, $|E⊥|$, plotted for comparison with the DFT calculations for $MoS_2$ (Fig. 1j–l) showing triangular symmetry due to the hexagonal lattice of $MoS_2$, and the absence of $|E⊥|$ in the center of each triangle corresponds to the location of the nucleus. In $WS_2$ (Fig. 1n), the larger atomic number of W (74) gives increased $|E⊥|$ compared to 2S (32), whereas in $MoS_2$ (Fig. 1k, l), $|E⊥|$ around Mo ($Z = 42$) is only just slightly higher than that around 2S due to the similar total scattering cross section. The $E⊥$ vector map (Fig. 1q, p) shows field lines pointing away from the nucleus with minima at the mid-point between Mo-2S bond and in the center of the hexagonal lattice.

**4D STEM of simple line vacancies**. We studied fluctuations of $E⊥$ within the 1D channels of S line vacancies formed in TMDs by electron beam irradiation (Fig. 2a, b). 1D vacancy channels contain metal−metal bonding sites due to loss of S atoms that are predicted to give rise to conduction channels within the 2D material. These metal−metal bonding channels will have different electrostatics compared to the bulk crystals due to changes in local bonding and electron sharing. The $E⊥$ at S point vacancies has also been characterized (supplementary figure 3.). Ultralong 1D line vacancies of 2S width (2SLV) are the most

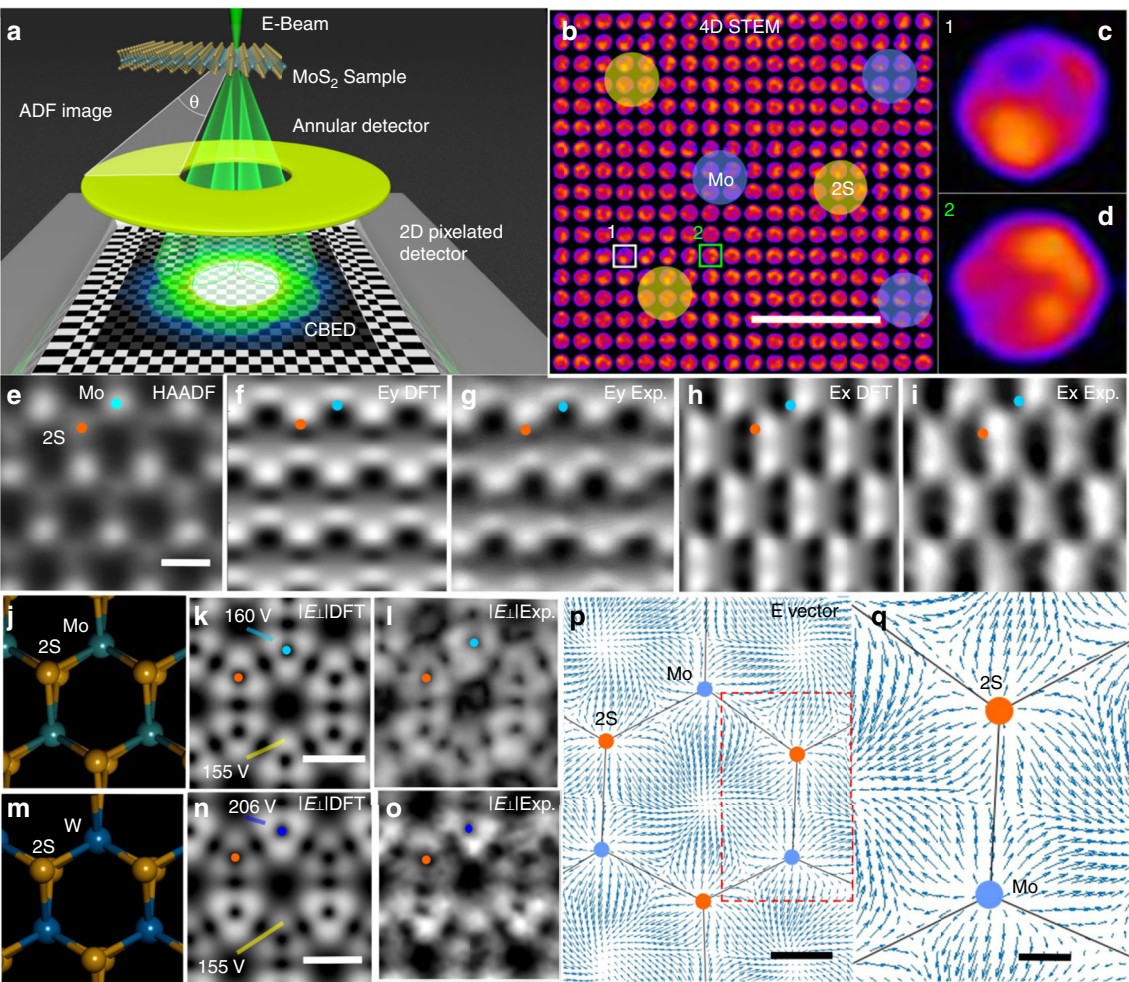

**Fig. 1** Atomic resolution 4D STEM of pristine $MoS_2$ and $WS_2$. **a** Schematic illustration of the 4D STEM geometry used. **b** 4D STEM data plotted as montage of CBED patterns, showing variations as a function of probe position relative to Mo and 2S atom positions. Scale bar indicates 0.2 nm. **c** CBED pattern from the position marked with the white box 1 in **b**, and **d** CBED pattern from the position marked with the green box in **b**. **e** HAADF-STEM image of $MoS_2$ taken using the ADF detector. Scale bar indicates 0.2 nm. **f** DFT-calculated $E\perp y$ field around $MoS_2$. **g** Experimental $E\perp y$ field around $MoS_2$ measured from $I_{comy}$, normalized and scaled to the range matching the DFT values. **h** DFT-calculated $E\perp x$ field around $MoS_2$. **i** Experimental $E\perp x$ field around $MoS_2$ measured from $I_{comx}$, scaled to the range matching the DFT values. **j** Atomic model of $MoS_2$ in a tilted projection to show 2S column. **k** DFT calculated $|E\perp|$ around $MoS_2$, according to the atomic model in (**j**). Scale bar indicates 0.2 nm. **l** Experimental $|E\perp|$ around $MoS_2$ measured from the $I_{com}$, scaled to the range matching the DFT data. **m** Atomic model of $WS_2$ in a tilted projection to show 2S column. **n** DFT calculated $|E\perp|$ around $WS_2$, according to the atomic model in **m**. Scale bar indicates 0.2 nm. **o** Experimental $|E\perp|$ around $WS_2$ measured from $I_{com}$, scaled to the range matching the DFT data. Orange dots in **e−n** indicate S atoms position, cyan dots represent Mo and blue dots W. **p** 2D map of the $E\perp$ vector, from the $I_{com}$ vector, in arrow representation (arrow size indicates the magnitude of vector) around the $MoS_2$ lattice. Scale bar indicates 0.1 nm. **q** Magnified view from the red dashed boxed area in **p** showing a high-resolution $E\perp$ vector plot around a single Mo atom and 2S column. Scale bar indicates 0.05 nm

common and simplest observed (Fig. 2b and supplementary figure 4). The tilt and in-plane bond compression make it hard to fully resolve this 2SVL using ADF-STEM imaging (Fig. 2c and supplementary figure 5). The ptychographic reconstructed phase (Fig. 2d) has higher resolution but is sensitive to height fluctuations[21,22]. The $|E\perp|$ map (Fig. 2e) has high sensitivity to local bonding coordination and can enable full structure determination of the 2SVL. The DFT calculations predicted a similar $|E\perp|$ image (Fig. 2f) based on the model in Fig. 2b. The vector map of $E\perp$ around the 1D channel is shown in supplementary figure 6. An atomic resolution image of the total charge is produced from the divergence of the $E\perp$ (Fig. 2g) revealing regions that are negative rich (blue) and positive rich (red). The regions with positive-rich charge are localized around the nuclei due to the protons, while the negative-rich regions are at the center of the hexagons and in the 2SVL 1D channels. The latter gives rise to a 1D atomically confined negative-rich channel

embedded within the 2D lattice, where S depletion is greatest. DFT calculations of the electron charge density maps (Fig. 2h, i) show a small increase in electron density in this area compared to the center of the hexagonal lattice (Fig. 2i). Figure 2i is the same image as Fig. 2h, but with the scale replotted to truncate the z-axis scale to 10 Å$^{-2}$, whereas in Fig. 2h it is 20 Å$^{-2}$. This rescaling helps to see the contrast from the 1D channel area. Similar results are obtained for the 2SVL in $WS_2$ monolayers (Fig. 2j−m), where the W−W bonding also provides the negative-rich 1D channels. The $E\perp$ images for $MoS_2$ and $WS_2$ both show distinct profiles for the metal atoms located in the 2SVL channel, which are shown in more detail in Fig. 3. DFT calculations of the in-gap states, supplementary figure 7, for the 2SVL are shown in Fig. 2n, o, with localization at the 1D channel in the same region. These results confirm the experimental imaging of 1D conduction channels that match the theoretically prediction location.

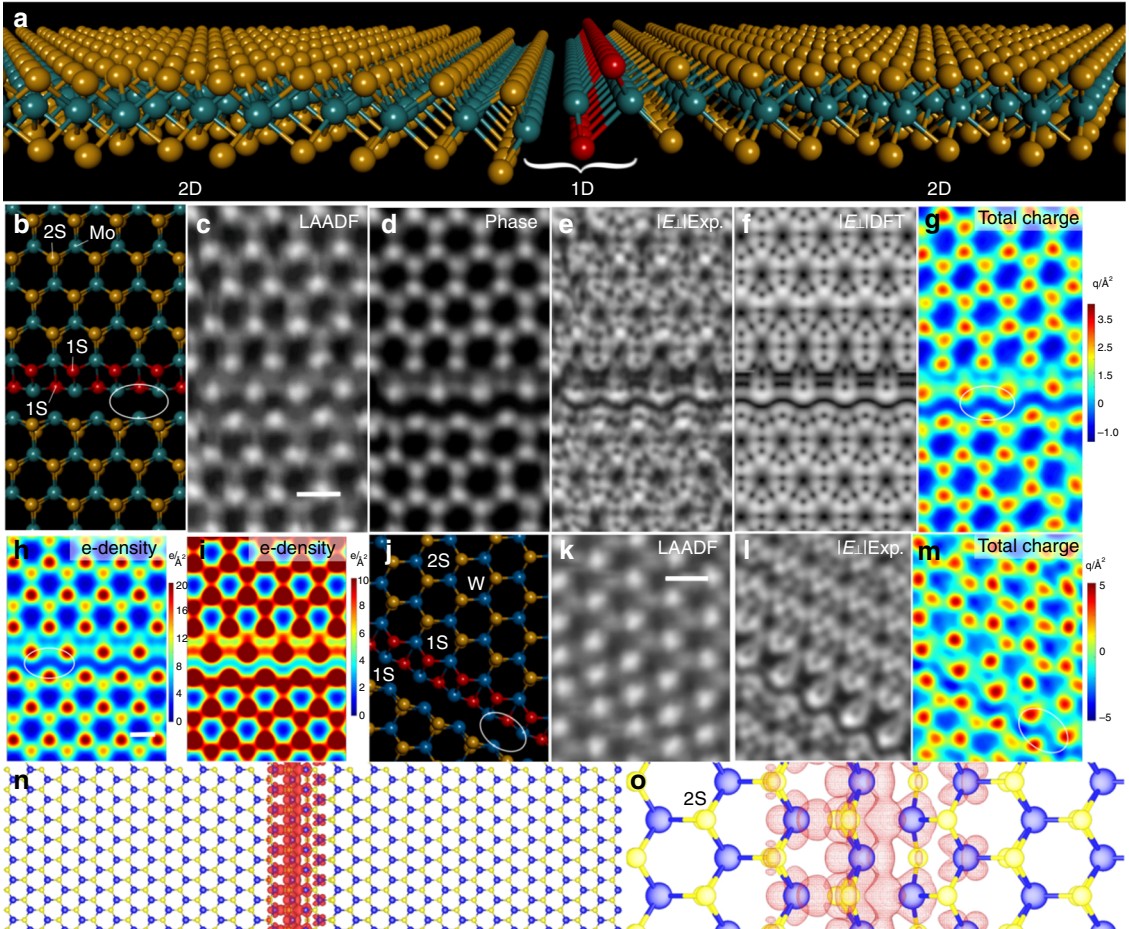

**Fig. 2** 4D STEM around 1D line defects in MoS$_2$ and WS$_2$. **a** Perspective view of an atomic model of the 1D vacancy line consisting of 2S missing atomic rows. **b** Atomic model of an MoS$_2$ line defect (2S), red indicates 1S and orange 2S. **c** LAADF image of a line defect area in MoS$_2$ reconstructed from 4D STEM data. Scale bar indicates 0.3 nm and is applicable for panels (**b**)−(**g**). **d** Phase map of a line defect area in MoS$_2$ from ptychography using 4D STEM data. **e** Experimental |$E_\perp$| map around a line defect in MoS$_2$: $I_{comx}$, normalized and scaled to match the DFT values. **f** DFT calculations of |$E_\perp$| around a line defect in MoS$_2$. **g** Total charge map ($q$/Å$^2$) around line defect in MoS$_2$ from 4D STEM data, scaled and normalized to match DFT range. **h** Electron charge density map (electrons/Å$^2$) around a line defect in MoS$_2$ calculated from DFT. Scale bar indicates 0.2 nm and is applicable for panel (**i**). **i** Map as in (**h**), scaled to emphasize the electron density in the Mo−Mo channel region. **j** Atomic model of line defect in WS$_2$ (2S form). Red indicates 1S and orange 2S. **k** LAADF image of a line defect in WS$_2$ reconstructed from 4D data. Scale bar indicates 0.3 nm and is applicable for panels (**j**)−(**m**). **l** |$E_\perp$| map around line defect in WS$_2$ from 4D STEM data: $I_{comx}$, normalized and scaled to the range to match the DFT values. **m** Total charge map ($q$/Å$^2$) around a line defect in WS$_2$, scaled and normalized to match DFT range. White ovals indicate metallic bonding region in 1D channel. **n** DFT calculated plot of the in-gap states (red) for WS$_2$ 2SV line defect. **o** Magnified view of the DFT calculated plot of in-gap states (red mesh) from (**n**)

The quantitative total charge density maps around the hexagonal unit of MoS$_2$ and WS$_2$ are examined in detail in Fig. 3. Figure 3a–e shows the 2S column has only slightly lower total charge than the Mo atom, but that W significantly heavier atomic mass shows substantial contrast compared to 2S sites, which is confirmed by DFT calculations. The |$E_\perp$| image and DFT calculations around the 1D channel region show agreement (Fig. 3g–h). The total charge maps around this region also show correlation between experiment (Fig. 3i) and DFT calculations (Fig. 3j). The strong blue region in Fig. 3g, h correlates to the Mo−Mo bonding region, Fig. 3f, where the electric field is significantly reduced in a 1D manner compared to the rest of the periodic lattice. This is more evident in the total charge maps in Fig. 3i, j, where the sign becomes negative, indicating electron-rich area in the 1D channel.

Similar agreements are also found in the WS$_2$ 1D channels (Fig. 4) where the electric field and total charge within the 1D channel associated with W−W bonding are distinctly different from the bulk lattice. The larger difference in the electric field

around W atoms compared to 2S results in more complex electric field patterns in Fig. 4a–c, compared to the case of MoS$_2$ in Fig. 3f–h. The total charge maps in Fig. 4d, e also contain the electron-rich 1D channel area, but larger variations in the values are seen along the channel area because of the difference in W and 2S sites. Line profiles along the 1D channel (Fig. 4f, g) show oscillations of the total charge in both experimental and DFT data, but still remaining electron rich.

**4D STEM of complex line vacancies.** Finally, we explore how the electric field maps and the total charge maps change when the width of the vacancy lines increases (Fig. 5). We found that by combining the $E_\perp$ 2D maps, total charge maps with the ADF-STEM and phase contrast images, it was possible to resolve the structure of these complex defective areas with a higher degree of confidence. The ADF-STEM image of the vacancy line in Fig. 5a is not easy to rapidly differentiate from the thinner vacancy line ADF-STEM image in Fig. 2k. However, the wider 3S line vacancy (3SVL) in WS$_2$

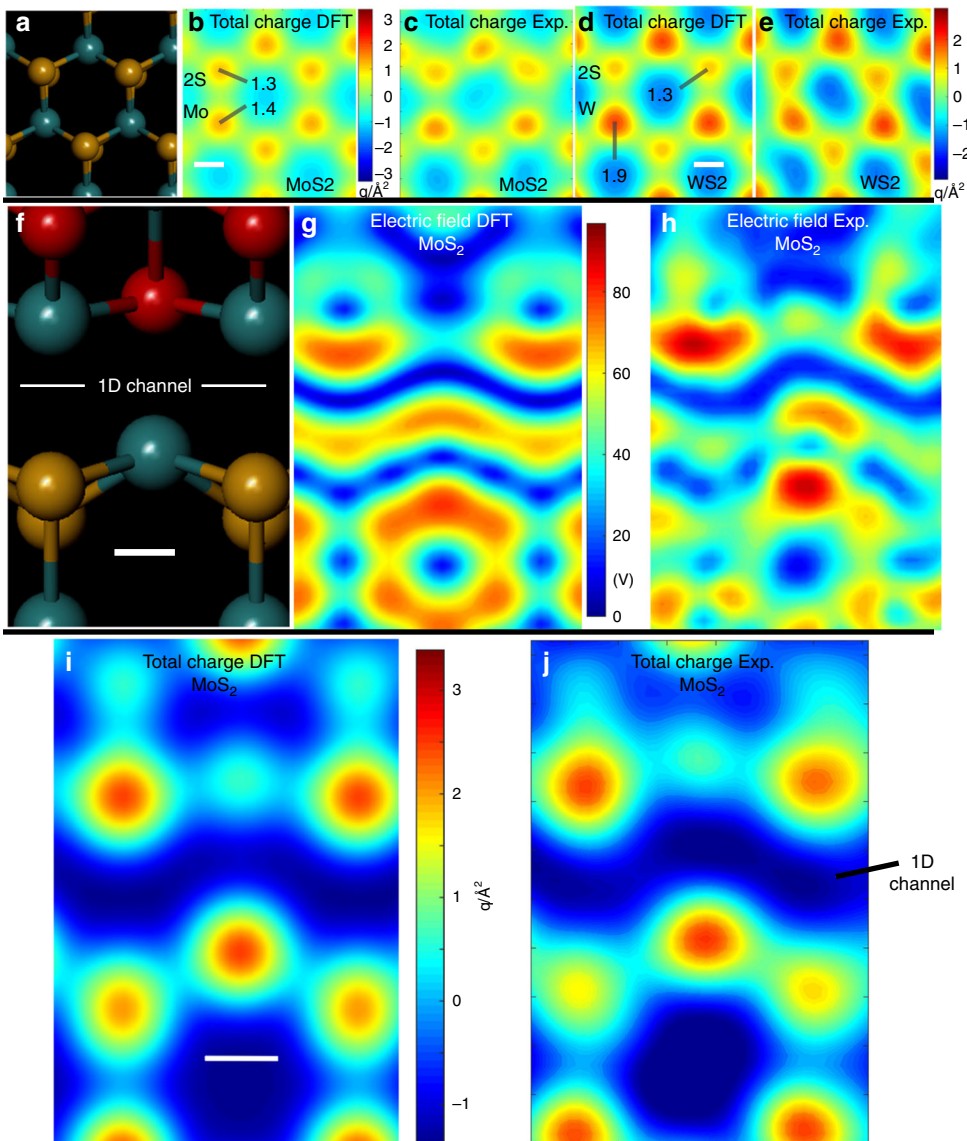

**Fig. 3** High-resolution analysis of electric field and total charge around pristine WS$_2$ and a 1D line vacancy in MoS$_2$. **a** Atomic model of MoS$_2$, (**b**) total charge map from DFT calculations for MoS$_2$ (scale bar indicates 0.1 nm and is applicable to panels (**a**–**e**)). Units are in elementary charge per Angstrom$^2$ and **c** experimental total charge map for MoS$_2$. **d** Total charge map from DFT calculations for WS$_2$ (scale bar indicates 0.1 nm and is applicable to panels (d) and **e** and **e** total experimental charge map of WS$_2$. **f** Atomic model of the 1D channel Mo−Mo bonding region in MoS$_2$ line defects (scale bar in panel (**f**) indicates 0.1 nm and is applicable to panels (**f**)−(**h**)). **g** |$E_\perp$| map calculated using DFT and **h** experimental |$E_\perp$| map for MoS$_2$. **i** Total charge map calculated from DFT for MoS$_2$ from the atomic model in panel (**f**) (scale bar indicates 0.1 nm and is applicable to panels (**i**) and **j** and **j** corresponding total experimental charge map

(Fig. 5a–e) with the $E_\perp$ 2D map in Fig. 5d shows ring patterns around the W atoms that are next to S vacancy sites, that are easy to differentiate compared to the pattern in Fig. 2l. As line vacancies get wider, the ADF-STEM images show complex contrast profiles (Fig. 5f–j, supplementary figure 8 and 9). Figure 5f is an ADF-STEM image taken with longer dwell time and consequently higher signal to noise and shows two line vacancies appear to be paired together with a gap in between. Identifying this complex defect region is challenging when relying purely upon a single ADF-STEM image. But the electric field images in Fig. 5i, j help to reveal the bonding at each atomic site. This is shown in higher magnification and detail in Fig. 6 for the case of MoS$_2$ double line defects.

A complex wide line defect in MoS$_2$ (Fig. 6) shows additional information in the |$E_\perp$| image (Fig. 6c) compared to the ADF-STEM (Fig. 6a) and reconstructed phase (Fig. 6b). ADF-STEM and reconstructed phase only provide an intensity for each

atomic position, whereas in the electric field image, both intensity and shape fluctuations occur. This is because it is influenced by both the total atomic number and the way it is bonded to its nearest neighbor. This is evident when comparing the electric fields around individual Mo atoms within the vacancy region to those Mo atoms in the pristine lattice, where the Mo atoms within the defective area have a different local electric field pattern (i.e. donut shaped). Under the low dose conditions used in the experiment, it is hard to differentiate 1S and 2S sites based on line profile analysis of ADF-STEM or phase images (Fig. 6h, k). However, a |$E_\perp$| image is sensitive to S vacancies and Mo bonding coordination and combined with ADF-STEM and phase data, an accurate description of the atomic structure can be deduced (Fig. 6e), and confirmed by the DFT-calculated |$E_\perp$| (Fig. 6g). See also supplementary figure 10 where the DFT-calculated electric field maps show distinctly different patterns for 1S and 2S sites.

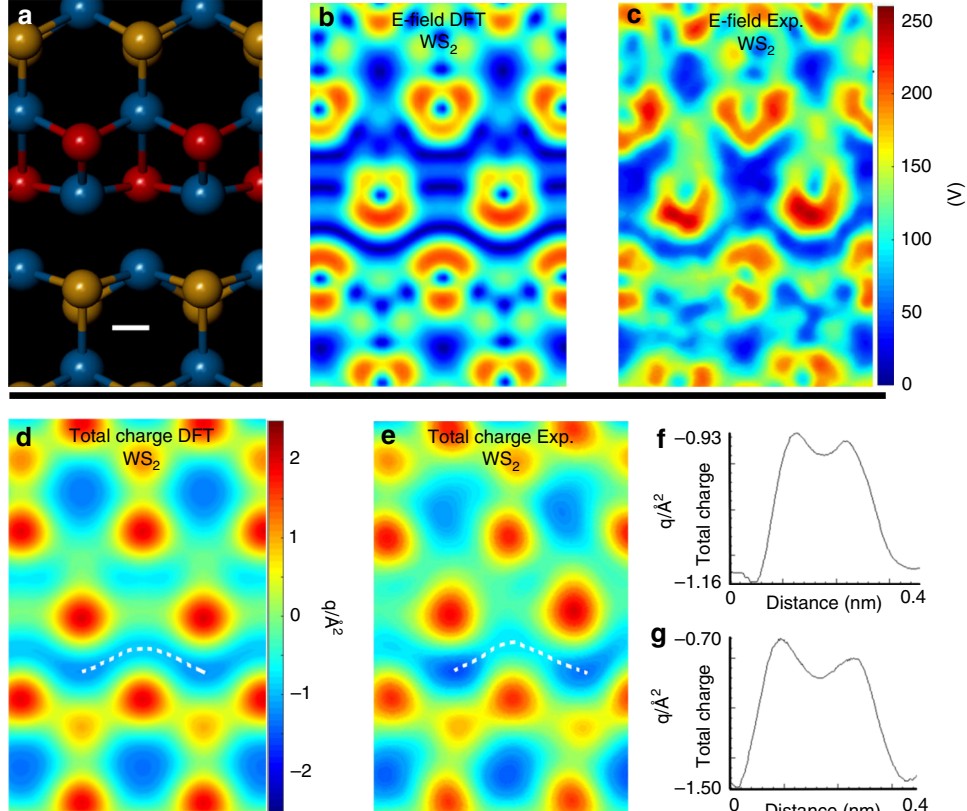

**Fig. 4** High-resolution analysis of electric field and total charge around a 1D line vacancy in WS$_2$. **a** Atomic model of the W−W bonding region at WS$_2$ line defects (scale bar indicates 0.1 nm and is applicable to panels (**a**–**e**)). **b** |$E_\perp$| map calculated using DFT and **c** experimental |$E_\perp$| map for WS$_2$. **d** Total charge map for WS$_2$ from the atomic model in panel (**a**), calculated from DFT. **e** Corresponding experimental total charge map. **f** Line profile taken along the white dotted line in **d**, from left to right, and **g** line profile taken along the white dotted line in **e**, from left to right

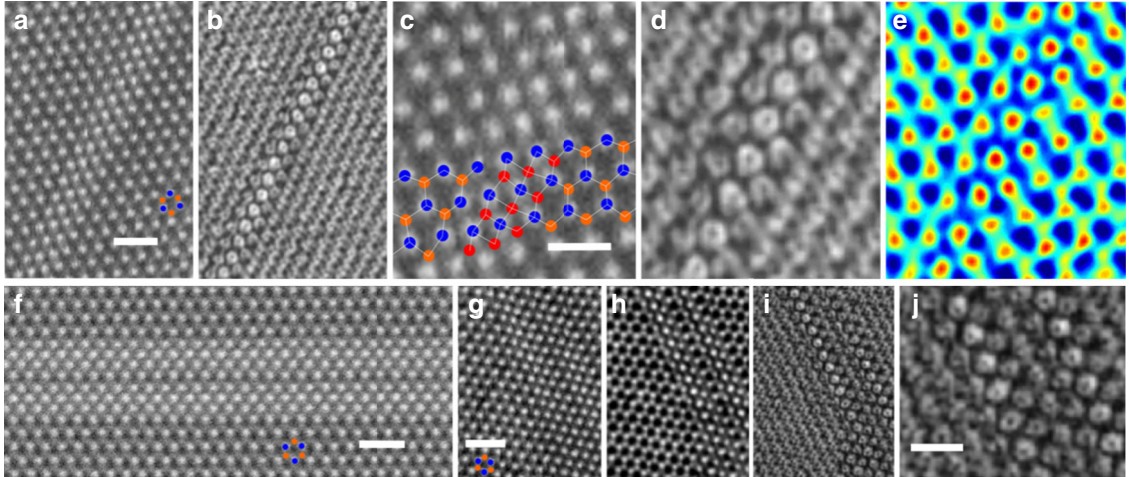

**Fig. 5** 4D STEM of larger width line vacancy channels in WS$_2$. **a** LAADF image of a 3S line defect in WS$_2$ reconstructed from the 4D STEM data. Scale bar indicates 1 nm and is applicable for panel (**b**). Blue spots indicate W and orange 2S positions. **b** |$E_\perp$| image reconstructed from 4D STEM data. **c** Magnified view of the ADF-STEM image reconstructed from 4D STEM data of a 3S line vacancy, S vacancy sites indicated by red dots. Blue dots mark W atom sites and orange dots 2S sites. Scale bar indicates 0.5 nm and is applicable to panels (**c**)−(**e**). **d** |$E_\perp$| image reconstructed from 4D STEM data. **e** Total charge map reconstructed from 4D STEM data. Color scaling is the same as that used in Fig. 2m. **f** High-resolution ADF-STEM image recorded using an annular detector of a complex wider line defect in WS$_2$. Scale bar indicates 1 nm. Blue spots indicate W atom sites and orange indicates 2S sites. **g** LAADF image of a complex line defect in WS$_2$ reconstructed from the 4D STEM data. Scale bar indicates 1 nm and is applicable to panels (**g**)−(**i**). Blue spots indicate W atom sites and orange indicates 2S positions. **h** Ptychographic phase of a complex line defect area in WS$_2$ reconstructed using the ePIE algorithm applied to 4D STEM data. **i** |$E_\perp$| map around a complex line defect in WS$_2$ reconstructed from 4D STEM data. **j** Magnified view of the |$E_\perp$| map in (**i**). Scale bar indicates 0.5 nm

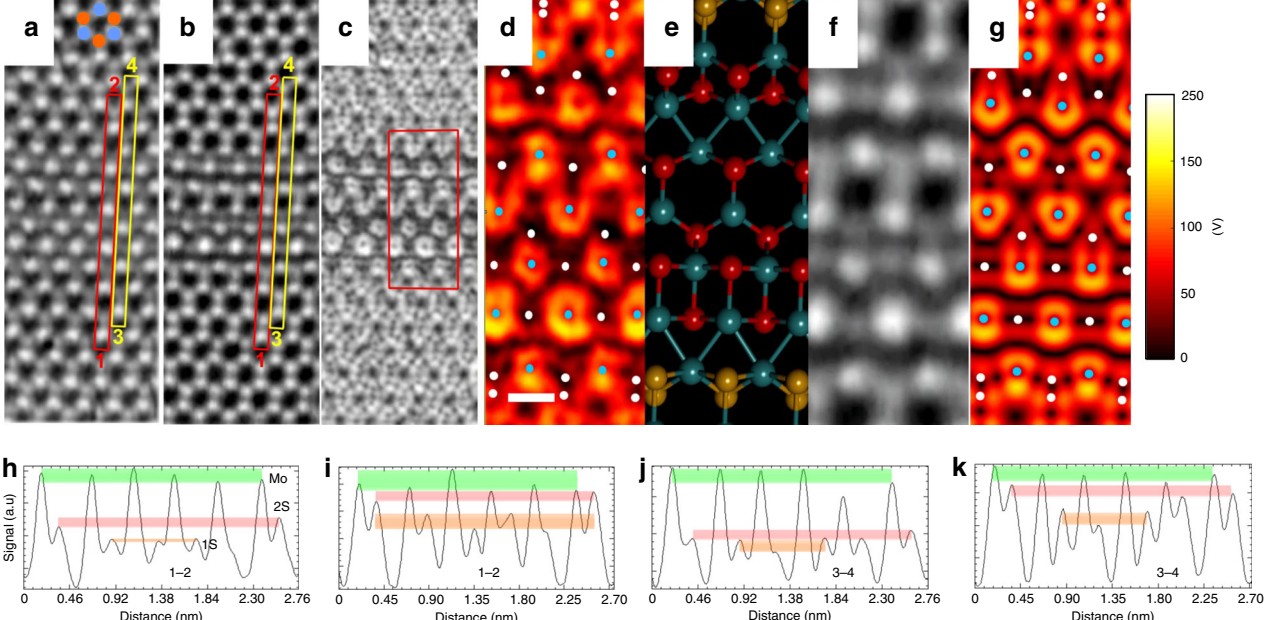

**Fig. 6** High-resolution 4D STEM of larger width line vacancy channels in $MoS_2$. **a** LAADF image of complex line defect in $MoS_2$ reconstructed from 4D STEM data. Scale bar indicates 0.5 nm. Cyan spots indicate Mo atom sites and orange indicates 2S positions. **b** Ptychographic phase of a complex line defect area in $MoS_2$ reconstructed from 4D STEM data. **c** $|E_\perp|$ map around complex line defect in $MoS_2$ reconstructed from 4D STEM data. **d** Magnified view of the red boxed area in **c**, plotted in color. Cyan dots indicate Mo atom sites and white dots indicate S atoms. Scale bar indicates 0.2 nm and is applicable to panels **d**−**g**. **e** Atomic model matching the structure in **d**. **f** Ptychographic phase map of the same area as in **d**. **g** DFT-calculated $|E_\perp|$ map, based on the atomic model in **e**, plotted in color. Cyan dots indicate Mo atom sites and white dots indicate S atoms. **h**−**k** Normalized line profiles taken from **h** red box in **a**, **i** yellow box in **a**, **j** red box in **b**, **k** yellow box in **b**. Green bands indicate the Mo level, red bands indicate the 2S level and orange band indicates the 1S level

## Discussion

Our results provided a quantitative comparison between electric fields and total charges in monolayer $MoS_2$ and $WS_2$ with experimental measurements from 4D STEM. A close match is found, but a scaling factor of 2 was needed for exact matching, which may be due to residual aberrations and further work is needed to understand this in more detail. We showed that 4D STEM can be used to directly locate electron-rich areas of 1D conduction channels in line vacancies, down to the metal−metal bonding area and revealing electron-rich sections that match the DFT predictions. Fluctuations of total charge along the 1D channels were predicted by theory and observed in the experiments. The electric field maps contain intensity fluctuations and complex local patterns around atoms that help to identify large defect structures when combined with the ADF-STEM and phase maps all acquired simultaneously by the 4D STEM approach. This approach should open up opportunities in the use of 4D STEM to map out electrostatic maps of molecules and thin materials.

## Methods

**TEM sample preparation**. $MoS_2$ and $WS_2$ monolayers were grown using chemical vapor deposition (CVD)[23]. An $SiO_2$ (300 nm)/Si substrate was used for growth and was sonicated in acetone, followed by an oxygen plasma treatment. CVD was carried out using 20 mg of $MoO_3$ powder (99.5%, Sigma-Aldrich) and 500 mg of S (99.5%, Sigma-Aldrich) as the precursor with Ar as the carrier gas under atmospheric pressure. A 500 sccm of Ar gas was used to purge the tubes for 30 min, followed by heating the S powder to ~180 °C for 10 min under an Ar flow rate of 150 sccm. The second furnace was heated at a rate of 40 °C min⁻¹ to 800 °C, whereas the $MoO_3$ powder reached a temperature of 300 °C. The reaction was conducted at 800 °C for 20 min with 10 sccm Ar. For $WS_2$, CVD was performed using the same system, but with 200 mg of $WO_3$ precursor and 300 mg of S powder, with a growth temperature of 1170 °C for 3 min. Samples were transferred using a PMMA support onto DENS in situ heating chips that contained slits in the thin SiN windows, cut by a focused ion beam. The PMMA was removed by acetone before TEM examination.

**4D STEM**. Scanning transmission electron microscopy was performed using an aberration- corrected JEOL ARM300CF equipped with a JEOL ETA corrector[24], operated at an accelerating voltage of 60 kV. The camera length was 6 cm, when imaging $MoS_2$, the aperture used was CL2-2 (40 μm), convergence semi-angle 39.1 mrad, beam current 48.5 pA; for $WS_2$, the aperture was CL2-3 (30 μm), convergence semi-angle 30.6 mrad, beam current 28.3 pA. Each 4D STEM data set contains 256 × 256 6-bit CBED patterns, dwell time in each CBED pattern (equals to a pixel in the final image) is 0.5−1 ms. Simultaneous HAADF detector was used, with inner collection angle of 111 mrad and outer collection angle of 223 mrad. The ultrafast 2D pixelated detector used for the 4D STEM work is the Merlin for EM Hybrid Pixel Detector (HPD), developed by Diamond Light Source and is built around the Medipix3 ASIC[5]. It combines direct detection of electrons with rapid readout in pixelated form for dynamic imaging. Noise-less detection of single-electron events and near ideal DQE and MTF detector responses down to 60 kV energies make this system ideal for our proposed work.

**High-temperature imaging up to 800 °C**. This was performed using a commercially available in situ heating holder from DENS Solutions (SH30-4M-FS). Heating the sample was achieved by passing a current through a platinum resistive coil imbedded in the TEM chip (DENS Solutions DENS-C-30). The resistance of the platinum coil was monitored in a four-point configuration, and the temperature was calculated using the Callendar−Van Dusen equation (with calibration constants provided by the manufacturer). Slits were fabricated in the $Si_3N_4$ membranes using focused ion beam milling before transferring the $MoS_2$. In situ heating to 800 °C was used to reduce surface contamination and to reduce hole opening from chemical etching. Long line defects are formed at this high temperature.

**Data processing**. Codes written in Matlab were used to process all 4D STEM data and DFT data.

Electric field maps were generated from the CBED pattern captured by the 2D pixelated detector by first measuring the intensity center of mass x, $I_{comx}$, and y, $I_{comy}$, components, which are proportional to the electric field components $E_x$ and $E_y$, respectively of the electric field vector perpendicular to the electron beam direction, $E_\perp$ given by Eq. 1.

$$E_\perp = E_x i + E_y j, \qquad (1)$$

where $i$ and $j$ are the unit vectors in the $x$ and $y$ directions and $E_x$ and $E_y$ are the magnitudes in the $x$ and $y$ directions. $|E_\perp|$ is the magnitude of the total electric field perpendicular to the electron beam.

For the 4D STEM data, we convert the center of mass shift (CoM) to moment transfer $\langle p_\perp \rangle$ by Eqs. 2 and 3, where $h$ is Planck's constant, and $\lambda$ is wavelength.

$$h \cdot \sin(\text{CoM}) = \langle p_\perp \rangle \cdot \lambda, \tag{2}$$

$$\Rightarrow \langle p_\perp \rangle = h \frac{\sin(\text{CoM})}{\lambda}. \tag{3}$$

Based on Ehrenfest's theorem, ref. [10] showed that the measured Electric field ($E_\perp$) is given by Eq. 4, where $e$ is elementary charge and $v$ is the velocity and $z$ the thickness.

$$E_\perp = -\langle p_\perp \rangle_\perp \frac{v}{ez}. \tag{4}$$

The electric field in units of (V) is then given by Eq. 5.

$$E_\perp \cdot z = -\langle p_\perp \rangle_\perp \frac{v}{e}. \tag{5}$$

The electric field maps from DFT presented in this work are integrated across $z$, i.e. $\int E_\perp(z).dz$ for DFT, and therefore in units of Volts to match Eq. 5. Electric field values from 4D STEM were generally half that of the DFT calculations, and therefore we applied a scaling factor of 2 to match the images. This is similar to the prior report of electric fields in graphene in ref. [15] being half the value of the simulated experiment. It is worth noting that our theoretical values come from first principles calculations, not from simulated 4D STEM experiments. The electric field mentioned here $E_\perp$ is probe convolved ($I$) with the intrinsic electric field ($E_{\perp R}$) by $E_\perp = (E_{\perp R} \otimes I)$. Smearing of DFT results is undertaken to achieve similar convolution.

The total charge ($\rho$) in units of elementary charge per Å$^2$, was calculated from Eq. 6, as described in detail in ref. [10], using electric field in units of (V). $\varepsilon_o$ is the vacuum permittivity.

$$\text{div}\,(E_\perp z) = \frac{\rho}{\varepsilon_0}. \tag{6}$$

Giving:

$$\rho = \varepsilon_0 \left( \frac{\partial E_x}{\partial x} + \frac{\partial E_y}{\partial y} \right), \tag{7}$$

where

$$E_x = -E_\perp \left( \frac{\text{CoM}\,x}{\text{CoM}\,r} \right) \tag{8}$$

$$E_y = -E_\perp \left( \frac{\text{CoM}\,y}{\text{CoM}\,r} \right). \tag{9}$$

Discrete gradient values in Eq. 7 were typically calculated over ranges of 3–5 pixels, depending on the magnification of the data to generate the total charge maps in Fig. 2g, m. The resultant images were scaled accordingly to match the range of the DFT calculations for comparison. The vector plot of the $E_\perp$ in Fig. 1p, q was produced using the quiver function in Matlab, using the $Ex$ and $Ey$ values.

**Phase reconstruction.** Ptychographic reconstructions were performed using the extended ptychographical iterative engine (ePIE) method developed by Maiden and Rodenburg[21], using the implementation described in ref. [22]. We implemented the ePIE algorithm in Matlab, where the STEM probe measured intensities were used to update the object wave in a random order using the expression given in ref. [22]. The reconstruction space was padded to reduce boundary artifacts, and a fixed number of iterations were used. Because the electron probe was determined to have very low residual coherent wave aberrations, we used an idealized probe wavefunction for the reconstruction. Microscope parameters were taken from the experimental metadata.

**Scaling factors and comparison of DFT with experimental data.** For quantitative comparison of DFT and experimental values we first smeared the DFT results to have spatial resolution similar to the experimental data, see supplementary note 1. The smearing is done in Fourier space and the degree of smearing for each data set was defined by sigma. See supplementary data for more details. For Fig. 1f, g, k, n, a sigma of 1.2 was used. For the total charge plots in Fig. 3b, d, and electric field in Fig. 3g a sigma of 1.6 was used. A Gaussian blur of 1 was used as well in real space to match the experimental data. For the total charge density in Fig. 3i, no Gaussian blur was used. For the electric field maps in Fig. 4b, a sigma of 1.2 was used and no Gaussian blur. For the total charge maps in Fig. 4d, a sigma of 1.6 was used with a Gaussian blur of 1. Experimental results were generally half the magnitude of the DFT calculations and a scaling factor was used for each comparison. The difference between the DFT and experimental data is mentioned before in refs. [15,18], and is likely due to residual aberrations or defocus as stated in

ref. [18]. In Fig. 1l and 1o, scaling factor of 2.3 was used, with a Gaussian blur of data being 1. In Fig. 2g and 2m a scaling factor of 2.4 was used with a Gaussian blur of 1. For Fig. 3c, e, h a Gaussian blur of 2 was used with a scaling factor of 2. For Fig. 3j, a Gaussian blur of 1 was used with a scaling factor of 2. For Fig. 4c, a Gaussian blur of 1 was used with a scaling factor of 2. For Fig. 4e, a Gaussian blur of 2 was used with a scaling factor of 2.

**Numerical methods.** The shifts for the center of mass of probing electron beam are imparted by the microscopic electric field generated by the charges in the suspended layered material. Here we assume the effects of the probing electrons on the layer itself can be ignored, and the microscopic electric field can be determined by the stationary ground state charge distributions. With the Ehrenfest's theorem, the probe electrons traveling at speed $v_z$ with initial momentum $\vec{p}_\perp = 0$ perpendicular to the normal direction of the atomic layer will acquire additional momentum after passing though the layer given by Eq. 10.

$$\Delta \vec{p}_\perp(\vec{r}) = -\frac{e}{v_z}\,dz \langle \vec{E}_\perp(z) \rangle_{\vec{r}}, \tag{10}$$

for an electron beam centered at $\vec{r} = (x, y)$ on the atomic layer. $\langle \vec{E}_\perp(z) \rangle_{\vec{r}}$ is the effective electric field on the probing beam wave-packet with a finite spread. We assumed a Gaussian density profile for the probing beam with a width $\sigma$ and the corresponding effective electric field is given by Eq. 11.

$$\langle \vec{E}_\perp(z) \rangle_{\vec{r}} = \left( \frac{1}{2\pi\sigma^2} \right) \int d^2 \vec{r}'\, e^{-|\vec{r}' - \vec{r}|^2/2\sigma^2} \vec{E}_\perp^M(\vec{r}', z). \tag{11}$$

The microscopic electric field $\vec{E}_\perp^M(\vec{r}, z)$ encodes the microscopic charge density distribution information in the layer. Given the charge density distribution $\rho^M(\vec{r}, z)$ from the electrons and the nuclei in the layer, the microscopic electric field $\vec{E}_\perp^M(\vec{r}, z)$ can be found by solving the Maxwell equation in the electrostatics. With the superposition and linearity properties of the equation, the $z$ integrated electric field can be obtained equivalently by solving for the planar charge $\rho(\vec{r}) = \int dz \rho^M(\vec{r}, z)$ at $z = 0$ plane with the dielectric constant $\varepsilon_o$ in space. We first expand the planar charge distribution into the Fourier components $\rho(\vec{r}) = \sum_{\vec{G}_i} \rho_{\vec{G}_i} e^{i\vec{G}_i \cdot \vec{r}}$

with $\vec{G}_i$ the reciprocal lattice vectors of the layer. Starting from the Poisson Eqs. 12

$$\varepsilon_0 \overrightarrow{\nabla}^2 \phi_{\vec{k}}(\vec{r}) = -\rho_{\vec{k}} e^{i\vec{k} \cdot \vec{r}} \delta(z) \tag{12}$$

For each component $\vec{k} = \vec{G}_i$ with charge $\rho_{\vec{k}} e^{i\vec{k} \cdot \vec{r}} \delta(z)$, the electric potential $\phi_{\vec{k}}(\vec{r})$ and field $\vec{E}_{\vec{k}}(\vec{r})$ are derived in Eqs. 13 and 14 respectively:

$$\phi_{\vec{k}}(\vec{r}) = \frac{\rho_{\vec{k}}}{2\varepsilon_0 |\vec{k}|} e^{-|\vec{k}||z|} e^{i\vec{k} \cdot \vec{r}} \tag{13}$$

$$\vec{E}_{\vec{k}}(\vec{r}) = \frac{-i\rho_{\vec{k}} \vec{k}}{2\varepsilon_0 |\vec{k}|} e^{-|\vec{k}||z|} e^{i\vec{k} \cdot \vec{r}} \tag{14}$$

With the $z$-integration of the electric field and the convolution from the probing beam Gaussian smearing, the total effective electric field is given by Eq. 15:

$$\tilde{E}^\sigma(\vec{r}) = \int dz \langle \vec{E}_\perp(z) \rangle_{\vec{r}} = \sum_{\vec{G}_i \neq 0} \frac{-i\rho_{\vec{G}_i} \vec{G}_i}{\varepsilon_0 |\vec{G}_i|^2} e^{i\vec{G}_i \cdot \vec{r} - |\vec{G}_i|^2/\sigma^2} \tag{15}$$

over non-zero $\vec{G}_i$ reciprocal lattice vectors ($\vec{G}_i = 0$ component is excluded from charge neutrality). The derivation can be generalized to a supercell geometry with defects. The results establish the connection between the measured effective electric field and the projected charge density distribution from the nucleus, core charges and valence electrons in the chemical bonding.

**Density functional theory calculation.** To simulate for the effective electric field and the charge density distribution in the layered compound compared with the experiment, we perform the ab initio density function theory (DFT) calculations for the pristine crystal and supercells with line defects. Electron charge density, being a core physical quantity in the formulation of DFT, can be handled differently depending on how core electrons are treated. In the all-electron-type DFT, the core electrons are treated as valence electrons that respond to the changes of the local environment, while in the pseudo-potential DFT the core electrons are frozen leaving only the valence electrons as the ones being optimized in the simulation. Though all-electron DFT is expected to be more accurate in treating all the electrons equally in the solids, the computation is also more demanding than the pseudo-potential DFT. In our numerical strategy, we perform both all-electron and pseudo-potential DFT for the pristine $MoS_2$ crystal that shows excellent agreement between the two. Having justified the validity of pseudo-potential DFT that assumes frozen core electrons, we further perform the pseudo-potential DFT

simulations for a supercell crystal with line defect. Below we give more details on numerical implementations.

For the all-electron DFT simulation, we employ a full-potential linearized augmented-plane wave (FP-LAPW) DFT calculation, implemented in the ELK code[25]. The local density approximation (LDA) exchange-correlation functional is used with a reciprocal $k$-grid size of $10 \times 10 \times 2$ for a pristine $MoS_2$ single layer crystal[26]. The muffin-tin radius between the atomic cores and the interstitial space is $R_{MT} = 1.29$ (1.09)Å for a molybdenum (sulfur) atom. The maximum angular momentum used for the augmented-plane wave is 10 and the plane wave basis for the interstitial region has a cutoff $8R_{MT}^{-1}$ (inverse of average Muffin-tin radius). The code supports the output of electron charge density and the microscopic electric field in space.

In our work, we also performed the pseudo-potential DFT implemented in Vienna Ab initio Simulation Package (VASP) to compute the relaxed equilibrium crystal structure and the corresponding ground state electron charge density distribution[27,28]. We use the PAW pseudo-potential formalism parametrized by PBE[29,30]. The $k$-grid sampling is $25 \times 25 \times 1$ for the pristine $1 \times 1$ unit cell and $1 \times 15 \times 1$ for the supercell with a line defect. The energy cutoff is 450 eV which sets the cutoff of this plane-wave-based pseudo-potential DFT code. The ground state charge density from the output is used to compute the effective electric field.

## Data availability
The data that support the findings of this study are available from the corresponding authors upon reasonable request.

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

## Acknowledgements
J.H.W. thanks support from the European Research Council (Grant no: 725258) S.F. and E.K. acknowledge support by the STC Center for Integrated Quantum Materials, NSF Grant No. DMR-1231319 and by ARO MURI Award No. W911NF-14-0247. The computations in this work were run on the Odyssey cluster supported by the FAS Division of Science, Research Computing Group at Harvard University. We thank Yang Lu and Si Zhou for providing some of the $MoS_2$ and $WS_2$ samples. We thank Diamond Light Source for access and support in use of the electron Physical Science Imaging Centre (Instrument E02 and proposal number 19045) that contributed to the results presented here. Work at the Molecular Foundry was supported by the Office of Science, Office of Basic Energy Sciences, of the U.S. Department of Energy under Contract No. DE-AC02-05CH11231.

## Author contributions
S.F. and E.K. contributed the DFT. Y.W., C.S.A, A.I.K. and J.H.W. contributed the 4D STEM measurements. Y.W. and J.H.W. contributed the 4D STEM data analysis. C.O. contributed the ptychographic phase reconstructions. Y.W., G.G.D.H., S.F. and J.H.W. contributed to the atomic models. J.H.W. contributed the manuscript writing with input from all co-authors.

## Additional information

**Competing interests:** The authors declare no competing interests.

