## [Peer Review File · Nature Communications]

Editorial Note: This manuscript has been previously reviewed at another journal that is not operating a transparent peer review scheme. This document only contains reviewer comments and rebuttal letters for versions considered at Nature Communications. Mentions of the other journal have been redacted.

REVIEWERS' COMMENTS:

Reviewer #1 (Remarks to the Author):

In this revised manuscript, the authors substantially modified their manuscript according to my previous comments, especially on the quantitiveness of their comparison between experiments and DFT calculations. Although still a scaling factor of ~ 2 is needed to achieve a perfect quantitative match between experimental 4D STEM data and DFT calculations, the authors successfully show the experiment and DFT show good quantitative matches. While authors reserve scaling factor issue to be resolved in future studies, the previous my issues on the quantitiveness is sufficiently documented in the revised manuscript.

I would like to ask the authors to clarify one thing. The authors pointed out that the 2SVL 1D channels have a slightly higher electron charge density than that at the center of hexagon as shown in Figure 2j. However, I am still wondering this is also true in the experiment. So, it may be better to show the intensity profiles across both the 1D channel and the center of hexagon and quantitatively compare them. This is probably important comparison for this manuscript because the author claims that the 2SVL 1D defects should be electron rich channels and can be directly imaged by this technique.

Reviewer #2 (Remarks to the Author):

In this revised manuscript, transferred from [redacted] to Nature Communications, the authors have by and large addressed all the comments received for their prior submission. All reviewers had suggested a transfer to [redacted] and the choice of Nature Communications seems perfectly suitable. Of note are far clearer explanations and comments about the match between theory and experiment on a common, quantified scale, in particular in the Supplementary information document. While these changes are most welcome, some of the figures have remained the same: a casual reader may not notice the factor of ~ 2 scaling necessary to match theoretical and experimental contrast without delving deeper in the text.

I would therefore recommend providing *for each panel* a quantitative color scale and/or an indication of the contrast extrema (even if this makes for busier figures – see some of the earlier comments suggesting focusing on fewer examples to unclutter the argument). Simply mentioning that things are scaled by “approximately 2” seems a bit of a cop out... especially when the exact numbers are available.

The data and processing remain otherwise excellent and with the larger number of figures and longer text permitted in Nature Communications, the manuscript feels less dense even if some figures could have been split further (figures 2 and 5 for instance).

I would therefore support the publication of this work in Nature Communications but would insist on the intensity or color scales being systematically provided for all panels prior to final acceptance, to make for a truly quantitative comparison – the relatively poor (or scaled) match does not detract from the study and if anything provides avenues for interesting further investigation.

Reviewer #3 (Remarks to the Author):

To my opinion, the authors have now carefully revised the paper and made all the improvements recommended by all three reviewers. I believe the paper is well suitable for Nature Communications in its present form.

REVIEWERS' COMMENTS:

Reviewer #1 (Remarks to the Author):

In this revised manuscript, the authors substantially modified their manuscript according to my previous comments, especially on the quantitiveness of their comparison between experiments and DFT calculations. Although still a scaling factor of ~ 2 is needed to achieve a perfect quantitative match between experimental 4D STEM data and DFT calculations, the authors successfully show the experiment and DFT show good quantitative matches. While authors reserve scaling factor issue to be resolved in future studies, the previous my issues on the quantitiveness is sufficiently documented in the revised manuscript.

I would like to ask the authors to clarify one thing. The authors pointed out that the 2SVL 1D channels have a slightly higher electron charge density than that at the center of hexagon as shown in Figure 2j. However, I am still wondering this is also true in the experiment. So, it may be better to show the intensity profiles across both the 1D channel and the center of hexagon and quantitatively compare them. This is probably important comparison for this manuscript because the author claims that the 2SVL 1D defects should be electron rich channels and can be directly imaged by this technique.

Our Response: The 2SVL 1D defects are shown to be electron rich in the total charge plots. This is sufficient to prove they are electron rich. To be a conduction channel doesn't mean they have to have total charge lower than the center of a hexagon, because the 1D channel has a continuum of states in 1D connection, where the center of hexagons is confined. There is not much difference in the total value of charge between hexagon centers and the 1D channel areas in the experimental data.

Reviewer #2 (Remarks to the Author):

In this revised manuscript, transferred from [redacted] to Nature Communications, the authors have by and large addressed all the comments received for their prior submission. All reviewers had suggested a transfer to [redacted] and the choice of Nature Communications seems perfectly suitable. Of note are far clearer explanations and comments about the match between theory and experiment on a common, quantified scale, in particular in the Supplementary information document. While these changes are most welcome, some of the figures have remained the same: a casual reader may not notice the factor of ~ 2 scaling necessary to match theoretical and experimental contrast without delving deeper in the text. I would therefore recommend providing *for each panel* a quantitative color scale and/or an indication of the contrast extrema (even if this makes for busier figures – see some of the earlier comments suggesting focusing on fewer examples to unclutter the argument). Simply mentioning that things are scaled by “approximately 2” seems a bit of a cop out... especially when the exact numbers are available.

Our response: We have included in the methods section the exact numbers used for scaling in each figure panel. We believe it is better to make the figures clear and easier to read for the broad audience of Nature Communications, to see the qualitative match of the features, with primarily the specialist only concerned with the exact matching of quantitative values. With the shift of the methods from supporting information to the main text, the reader no longer needs to go to SI to find this information, as it is all within the main text of the manuscript.

The data and processing remain otherwise excellent and with the larger number of figures and longer text permitted in Nature Communications, the manuscript feels less dense even if some figures could have been split further (figures 2 and 5 for instance).

I would therefore support the publication of this work in Nature Communications but would insist on the intensity or color scales being systematically provided for all panels prior to final

acceptance, to make for a truly quantitative comparison – the relatively poor (or scaled) match does not detract from the study and if anything provides avenues for interesting further investigation.

Our Response: We are pleased with the positive response of the reviewer

Reviewer #3 (Remarks to the Author):

To my opinion, the authors have now carefully revised the paper and made all the improvements recommended by all three reviewers. I believe the paper is well suitable for Nature Communications in its present form.

Our Response: We are pleased with the positive response of the reviewer